**METHOD**

# vamos: variable-number tandem repeats annotation using efficient motif sets

Jingwen Ren[1†], Bida Gu[1†] and Mark J. P. Chaisson[1*]

†Jingwen Ren and Bida Gu contributed equally to this work.

*Correspondence:
mchaisso@usc.edu

[1] Department of Quantitative and Computational Biology, University of Southern California, Los Angeles, US

**Abstract**

Roughly 3% of the human genome is composed of variable-number tandem repeats (VNTRs): arrays of motifs at least six bases. These loci are highly polymorphic, yet current approaches that define and merge variants based on alignment breakpoints do not capture their full diversity. Here we present a method vamos: *V*NTR *A*nnotation using efficient *Mo*tif *Se*ts that instead annotates VNTR using repeat composition under different levels of motif diversity. Using vamos we estimate 7.4–16.7 alleles per locus when applied to 74 haplotype-resolved human assemblies, compared to breakpoint-based approaches that estimate 4.0–5.5 alleles per locus.

**Keywords:** Variable-number tandem repeats, Long-read sequencing, Motif composition

## Background

Variable number-tandem repeats (VNTRs) are a class of repetitive DNA composed of short DNA sequences called motifs repeated many times in tandem. By convention, VNTRs are composed of motifs at least six bases; shorter motifs are classified as short tandem repeats (STRs). The repetitive nature of these sequences primes them for mutations through strand slippage during replication and unequal crossing over that increase or decrease motif copy number or introduce mutations of motifs [1]. Variation of VNTR sequences has been found to impact physiology and cellular function. Disease studies have found associations of VNTR length or composition with diabetes [2], schizophrenia [3], and Alzheimer's [4]. Additionally, methods developed to analyze VNTR variation using high-throughput short read sequencing data found widespread association between VNTR length and gene expression [5–7]. Finally, variation directly in coding sequences are found to be linked with human traits including height and hair patterns [8, 9].

The overall knowledge of genetic diversity in these sequences lags behind non-repetitive DNA, primarily due to difficulties in genotyping VNTRs using short read data. The hypervariability of repeat counts and repeat composition among individuals

makes it challenging to interpret VNTR variation from read alignments due to alignment degeneracy and reference bias. As a consequence, STR and VNTR sequences have been masked in many large scale short read studies using low complexity filters [10, 11]. Specific methods have been written to analyze repeat unit variation in STRs, however these methods achieve precise counts of repeat units only for loci shorter than the read length [12] or insert-size [13], or for specific repeat patterns [14]. Methods developed to genotype VNTR variation using short-reads, only provide an estimate of motif count based on read depth inferred from genomic alignments [7], hidden Markov models [5, 15], and alignment to pangenomes [6].

Contrary to short-read sequencing, variation of VNTR loci is routinely resolved using long-read sequencing (LRS) [16–20] and assemblies [21, 22] based on the property that long read alignments or assemblies span across most VNTR loci. The initial long-read assembly studies found that insertions and deletions greater than 50 bases (structural variation) in humans are enriched in VNTR loci [21, 23]. More recently it was found that over 61% of structural variants discovered in 32 haplotype-resolved assemblies produced by the Human Genome Structural Variation Consortium [22] are inside VNTRs. The loci that harbor these structural variants are highly polymorphic. An analysis of ∼30,000 VNTRs in these assemblies (corresponding to loci studied in [22]) shows ∼9 alleles per locus, when considering each distinct sequence as an allele. This analysis with recent long-read data in human agrees with the long-studied observation that VNTR mutation rate in prokaryotes is up to six orders of magnitude greater than single-nucleotide polymorphisms [24, 25].

Variation is discovered using long-reads or their assemblies by mapping reads/assemblies to genomes using methods that allow for large insertions and deletions [26, 27], and recording variants by their position along with the number of bases gained or lost directly from alignments [17, 18]. When discovering variation from unassembled reads, variant signatures from separate reads are combined into individual calls allowing for inexact matching of variants to account for alignment differences driven by sequencing error [16–20]. Similarly, when combining variation discovered from multiple individuals into population-scale databases, variants in different samples are merged into distinct alleles depending on similarity of variant calls [20–22].

In both instances, merging variant signatures in tandem repeat loci is challenging in repetitive DNA because gaps in the reads/assemblies may be placed in different locations, depending on the alignment algorithm and gap penalties used [21]. The variant merging heuristics across haplotypes can collapse different mutations, resulting in a reduced measure of diversity of these sequences. Finally, recording variants strictly as gains and losses of DNA masks polymorphisms in repeat compositions, thus does not reveal the full extent of sequence mutations in VNTRs. For example, variation in an intronic VNTR of *CACNA1C* is associated with schizophrenia by repeat composition, rather than length [3], and pangenome-analysis of VNTR composition cis-association expression of 174 genes with changes in VNTR composition [28]. A recent study of 3622 Icelandic individuals sequenced with LRS used a combination of pre-filtering and clique-based clustering to provide a finer separation of variant alleles for VNTR sequences that associate with height, atrial fibrillation, and recombination [9]. As larger scale long-read sequencing studies become available, a more effective genotyping method suitable for

long reads may help shed light on the role of VNTRs in human genetics, since LRS and their assemblies has the potential to resolve large and complex VNTR alleles.

Rather than building a catalog of variation from differences in alignments, a more accurate approach to describe VNTR variation is to annotate their motif composition. By comparing tandem repeat sequences according to repeat units, the full diversity of a population or study cohort may be revealed as both differences in repeat length and composition. One naive approach to creating a motif database for each VNTR locus is to compile a non-redundant set of repeat motifs identified by a repeat discovery tool such as Tandem Repeats Finder (TRF) [29], applied to a reference set of VNTR sequences. However, this method may generate an extensive list that includes many rare motifs, which may obscure the pattern of repetition. For example, the sequence `ACGGTACGG TACCGTACGT` may be decomposed into `[ACGGT, ACGGT, ACCGT, ACGT]`; however, $(ACGGT)_4$ (4 repetitions) is more concise and has a low divergence from the original sequence. Furthermore, annotation by TRF across multiple samples may have inconsistent periodicity and starting frames. Thus we define an efficient motif set as a subset of original motifs consistently defined across a reference panel, in which rare motifs are replaced by more common ones while maintaining a bounded total replacement cost.

To find such efficient motif sets, we have developed a toolkit, VNTR Annotation Using Efficient Motifs Set (vamos) that finds efficient motif sets using a reference panel of diversity genomes. We have integrated the StringDecomposer algorithm [30] into vamos to annotate new genomes sequenced from aligned LRS reads or their assemblies using efficient motif sets. We generated an efficient motif set for VNTR loci from 148 haplotype-resolved assemblies sequenced with LRS by the Human Genome Structural Variant Consortium (HGSVC) [22] and the Human Pangenome Reference Consortium (HPRC) [31] and under three levels of divergence, as well as the original motifs. As a proof of concept, we used vamos to create a combined VNTR callset across the HGSVC and HPRC assemblies to quantify diversity of VNTR sequences, and compared this to the diversity measured by a separate approach that combines calls based on merging similar variants). We additionally evaluated the application of vamos to unassembled long read data.

## Results

### Efficient motif discovery for a reference assembly panel

Efficient motif set discovery and allele annotation were ran on the 148 HPRC and HGSVC assemblies. After preprocessing, an average of $360,864 \pm 6072$ annotated VNTR loci were retained per haplotype. The union of annotations of all 148 assemblies yielded a final set of 390,115 loci, of which only 15,514 were trivial repeats with a single motif. On average, there were $8.97 \pm 26.57$ motifs per locus.

The filtered motif sets were used as input for efficient motif discovery. To examine the motif diversity at varying levels of compression and population sampling, we generated efficient motif sets for an increasing number of reference assemblies, with $q$ values of 0.1, 0.2, and 0.3. Examples of VNTR decompositions using all observed motifs compared to efficient set of motifs annotations are shown in Fig. 1; the *ACAN* exonic VNTR found to be associated with height [8, 9], and an intronic VNTR in *WDR7* found to be a modifier of amyotrophic lateral sclerosis [32].

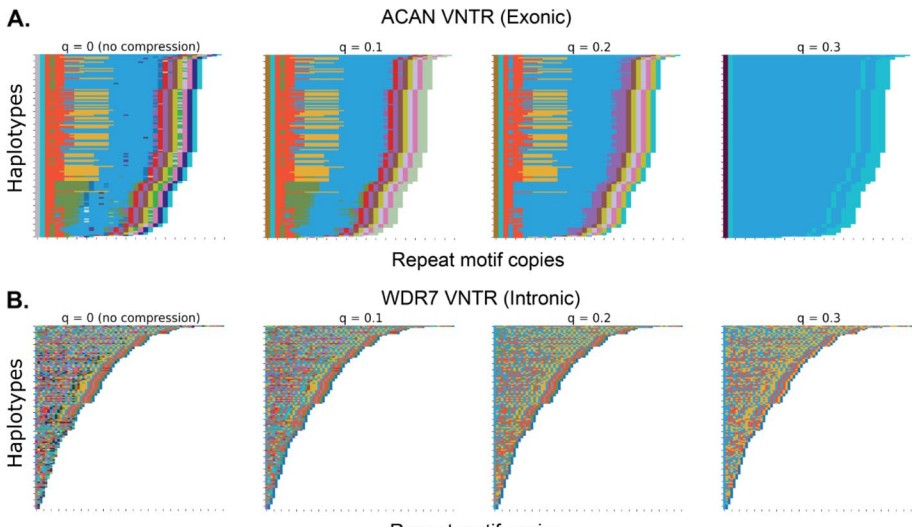

**Fig. 1** Visualization of allele structure change with four levels of $q$ ($q = 0, 0.1, 0.2, 0.3$). $q = 0$ means no compression on the original motif database. With $q$ growing larger, only truly representative motifs are selected and rare/contamination motifs possibly resulting from sequence error disappear and the allele's domain structure starts to grow more clear. The ACAN and WDR7 VNTR sequences from 148 HGSVC and HPRC assemblies are annotated into a string of efficient motifs selected at each level of $q$. Repeat motifs are color-coded and each individual annotation is plotted as a series of colors (row). Repeat motif unit is shown on *x*-axis. **A.** *ACAN* (chr15:88855422–88857301 on GRCh38) repeat alleles ranked by decreased length. **B.** *WDR7* (chr18:57024494–57024955 on GRCh38) repeat alleles ranked by decreased length

To assess the global diversity of VNTR motifs, we used genome-wide total motif count to measure the change in efficient motifs as more assemblies were added to the discovery panel from one to 148 assemblies. As expected, the number of the original motifs ($q = 0$) increased as genomes were added (Fig. 2), reflecting the inclusion of rare motifs with each additional assembly. The number of efficient motifs also increased as more assemblies are included, with total diversity showing asymptotic leveling at 148 genomes. Specifically, after inclusion of 30 assemblies, $\sim 99.07\%$ efficient motifs were selected compared to the entire set when 148 assemblies were incorporated with a $q$ value of 0.1 (Fig. 2).

On average, the efficient motif set size per VNTR locus ranges from 4.36 at $q = 0.1$ to 2.6 at $q = 0.3$ when 148 assemblies are included, while the original motif set size averages 8.97 per VNTR locus (Table 1). The mean compression ratio of motifs (efficient / original motif size) ranges from 0.78 to 0.63 ($q = 0.1 - 0.3$) (Table 1).

One possible approach to compile an efficient motif set is a greedy approach, which selects the most frequently occurring motifs at each locus. To compare the performance of efficient motif set selected by vamos algorithm and the greedy method, we generated efficient motifs with $q$ value of 0.1 by vamos and motif sets of the same size using the greedy method. Both motif sets were used to annotate a total of 136,748 VNTR loci for 148 HGSVC and HPRC haplotype-assemblies, excluding homogeneous loci with less than five original motifs. To evaluate the quality of the annotations, we computed the edit distance between the nucleotide sequences translated from the annotated string of motifs and the assembled raw sequences. We defined an annotation to be superior to another if its edit distance to the true VNTR sequence is less than 80% of the

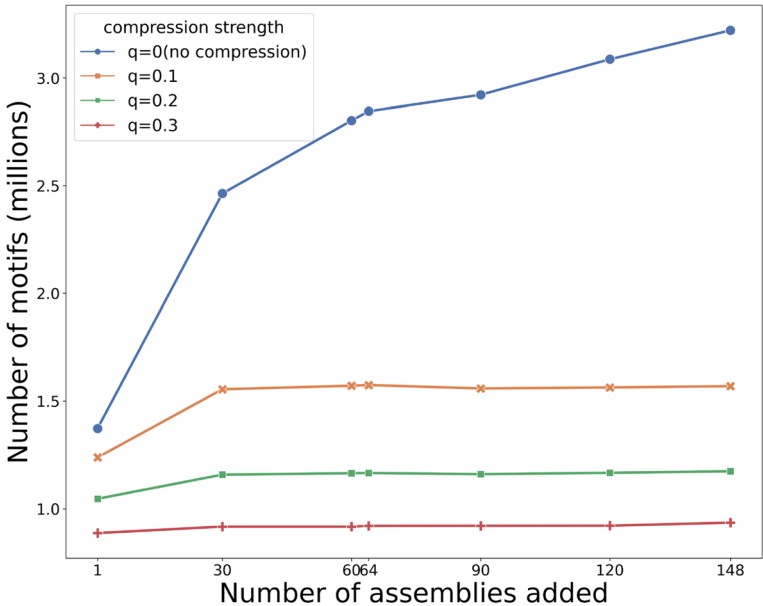

**Fig. 2** Diversity of efficient motif sets. Diversity of efficient motif sets under four levels of compression ($q = 0, 0.1, 0.2, 0.3$) as more assemblies are incorporated, measured by genome-wide number of efficient motifs. The curve of $q = 0$ reflects the number of original motifs. As expected, the number of the original motifs increases as genomes are added, due to the inclusion of rare motifs. The efficient motif set size increases as more assemblies are included, with total diversity in this category showing asymptotic leveling at 30 genomes. As $q$ grows larger, more compression is posed on the original motifs, resulting in less efficient motifs

**Table 1** The mean efficient motif set size and compression ratio under three levels of $q$ (0.1,0.2,0.3), when 148 haploid assemblies are incorporated

| $q$ | 0.1 | 0.2 | 0.3 |
|---|---|---|---|
| Mean efficient motif set size | 4.36 | 3.26 | 2.60 |
| Mean compression ratio | 0.78 | 0.69 | 0.63 |

**Table 2** Comparison between vamos and greedy method. The quality of an annotation was measured calculating the edit distance between the nucleotide sequences translated from the annotated string of motifs and the original assembled sequences. An annotation is significantly better if the corresponding edit distance is less than 80% of that of the other

| | Average number of loci per assembly |
|---|---|
| vamos < greedy | 15,170 |
| vamos <80%×greedy | 8097 |
| greedy < vamos | 13,479 |
| greedy <80%×vamos | 6308 |

corresponding edit distance of the other sequence. Under this metric vamos outperforms the greedy approach on $6.3 \pm 0.1\%$ (8097), which is roughly 1.3 times the number of loci where the greedy approach is superior (Table 2). One advantage of using efficient

motifs over the greedy approach is that it takes into account the cost of replacing each original motif, resulting in not only highly frequent motifs being reserved but also less frequent ones with long length and dissimilarity from other motifs. For instance, suppose a locus has three motifs `AAAG`, `AAAC`, and `TGTGACCTGCAC` with counts 10, 3 and 1. The greedy approach picks top two frequent motifs `AAAG` and `AAAC`, whereas vamos selects `AAAG`, `TGTGACCTGCAC`. By including the rare motif `TGTGACCTGCAC`, vamos efficient motifs allow for representation of more diverse sequences that include this motif. Another advantage of vamos is the automatic decision of the number of efficient motifs to select at each locus, which benefits from the ILP formulation and the association with a total replacement cost upper bound $\Delta$ that can adapt to the complexity of original motif set for each locus.

### Allelic diversity in 148 haplotype-resolved de novo assemblies

Because previous studies using long-read sequencing and assembly to quantify human diversity using methods that merged separate distinct alleles into the same variant [22], we sought to use vamos to quantify human diversity in VNTR sequences with motif-resolution in the HGSVC and HPRC haplotype-resolved assemblies. Each assembly was annotated independently using the original motif set and efficient motif sets defined by $q = 0.1$. The average number of alleles per locus under the original motif set was 4.8 (7.8 excluding constant loci), and under the efficient motif set was 4.1 (6.4 excluding constant loci) (Table 3, Fig. 3). Combining the HGSVC and HPRC data set resulted in about 26–33% more alleles than each individual set for $q=0$, and 21–33% more alleles with q=0.1, as evidence of the high variability of VNTRs. The number of different alleles correlates with the average motif count using both the original ($r^2 = 0.21, p < 2.2 \times 10^{-16}$) and efficient ($r^2 = 0.21, p < 2.2 \times 10^{-16}$) motif sets. In contrast, there were 3.3 alleles per locus when comparing by exact length, and 5.4 alleles per locus when comparing by exact sequence.

To measure how different alleles are, we also grouped together annotations if their edit distance with respect to the motif composition was up to 0–3 motifs (Table 3). When considering annotations using the original motif set, there was a 46% reduction in the average number of alleles per locus when grouping alleles that differ by up to two motifs by an edit distance allowing for insertion and deletion in addition to motif mismatch. Similarly, there was a 44% reduction in the average number of alleles in the efficient motif set under a similar grouping by edit distance. Because the annotation by efficient

**Table 3** Average number of alleles per locus when all 58 HGSVC and 90 HPRC haplotypes were annotated using the original and efficient set of motifs. Annotations were grouped into one allele if their edit distance with respect to the motif composition is 0–3 motifs. Statistics calculated by excluding constant loci that have only one allele measured by 0 edit distance under the original set are shown in brackets

| Motif set | Edit distance | | | |
| --- | --- | --- | --- | --- |
|  | 0 | 1 | 2 | 3 |
| Original set ($q = 0$) | 4.8 (7.8) | 3.4 (5.2) | 2.6 (3.8) | 2.2 (3.1) |
| Efficient set ($q = 0.1$) | 4.1 (6.4) | 2.9 (4.3) | 2.3 (3.3) | 2.0 (2.7) |

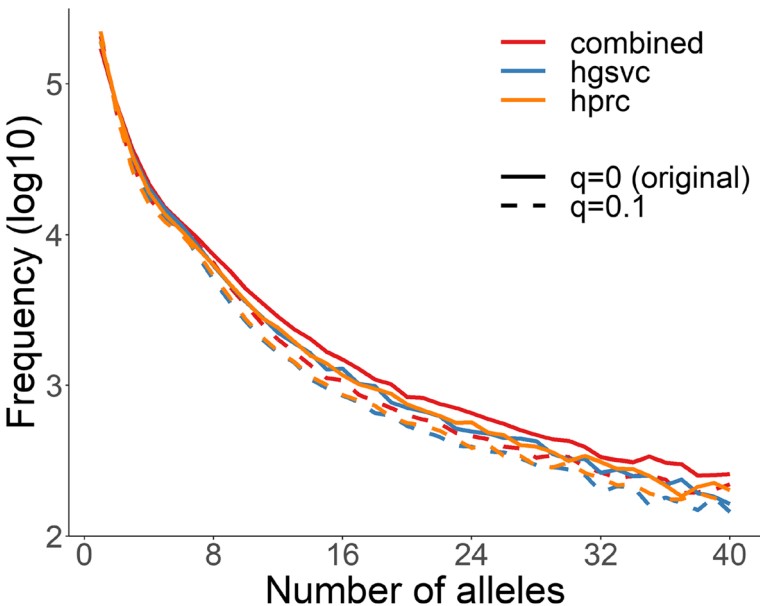

**Fig. 3** Distribution of number of VNTR alleles for 58 HGSVC and 90 HPRC haplotypes. VNTR sequences were annotated by *vamos –contig* using both efficient motifs ($q = 0.1$) and original motifs ($q = 0$). Alleles from the call set were counted by the annotation strings

motif sets already have a reduced diversity due to motif compression, the similar relative reduction by edit distance in both annotations indicates relatively few motif mismatches and that copy number variation instead accounts for the majority of allelic variation, as expected by the slippage mutational mechanism of tandem repeat sequences.

### Annotating diverged populations

We used simulation analysis to evaluate annotations of new genomes not included in the reference assemblies using simulated VNTR sequences so that the ground truth motifs are known. VNTR sequences were simulated based on 50,000 randomly selected loci described in Section "Generating an original motif set" and the corresponding *replacement annotations* described in Section "Efficient motif selection" were compared with the vamos annotations using StringDecomposer. Motif sets with a single motif were excluded from simulation since variations introduced by simulation may largely be concealed by the motif homogeneity in such cases. From each of the 50,000 selected motif sets, VNTR sequences were generated by sampling and concatenating 50–100 motifs according to their underlying frequencies in the reference assemblies.

Population diversity where sequences have motifs not reflected in the original or efficient motif sets was simulated by adding random mutations into the simulated VNTR sequences. Specifically, four settings for mutation rate: a 1% and 2% mutation rate sampled uniformly from single base substitution, deletion, or duplication combined with an optional 1% rate of insertions 2–4 bases (Fig. 4).

The annotations were assessed using simulated assemblies (error-free), as well as 30-fold coverage of reads from HiFi using pbsim [33] with average 98.1% accuracy and ONT using alchemy2 (distributed with lra [27] and averaging 88.8% accuracy). Similar to the analysis of aligned reads, the accuracy of the annotation was impaired by sequencing

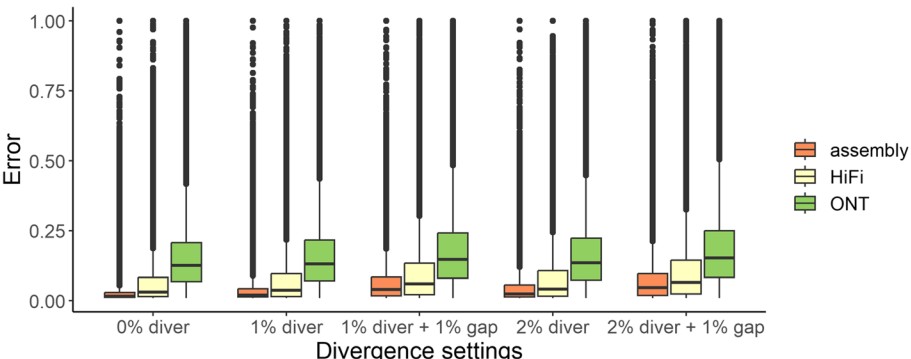

**Fig. 4** Error of vamos annotation in simulated data for assemblies, HiFi, and ONT reads under five divergence settings. The ground truth annotation is defined by replacing the original simulated motifs by their counterpart efficient motifs for $q = 0.1$. The error is calculated by dividing the edit distance on motifs between the vamos annotation and ground truth to the number of motifs repeated in the ground truth. The divergence and gap percentages refer to the number of base mutation and indel spaces added per hundred bases of the locus

error (Fig. 4). The addition of population diversity through sequence divergence further decreased the annotation accuracy. Overall, vamos exhibited robust performance on simulated assemblies and high quality sequencing reads ($\geq 98\%$ accuracy). The baseline error rate for annotating sequences without divergence was 6.1% for PacBio HiFi and 15.6% for ONT. When there was sequence divergence through mutation and indels, the error rate only rose modestly, from 6.6 to 10.0% for PacBio HiFi and 15.7 to 18.9% for ONT, indicating that relatively large differences in sequence variation may still have similar motif annotations in the efficient motif set. As technologies improve the relative differences between approaches are likely to decrease. The instances where there was a high divergence between the simulated motifs and the motif decomposition were typically due to long motifs in the original motif set being annotated by multiple shorter motifs in the efficient motif set.

**Analysis of aligned long-reads**

Because studies may elect to perform low-coverage sequencing to increase sample size, we analyzed long-read data of NA24385 sampled at $10–30\times$ for both HiFi and ONT platforms to investigate the accuracy of VNTR annotation on real mapped reads. Since analysis of assembly data gave the best overall performance by simulation, we compared annotations from reads data to those from the HPRC NA24385 assembly (Fig. 5). Regions where reads could not be phased, but still covered a VNTR locus were annotated with homozygous calls. The higher sequencing coverage leads to better phasing and construction of more accurate consensus sequences for improved overall performance. Increasing sequencing depth from $10\times$ to $20\times$ greatly decreased the number of uncovered loci for both platforms, yet such benefit was not as significant when sequencing depth was further increased from $20\times$ to $30\times$. Under the sequencing depth of $30\times$, the percentage of VNTR loci with over 80% of the covered reads successfully phased was 78% and 71% for the HiFi and ONT data, respectively. However, although both platforms had over 97% of the VNTR loci covered at $10\times$ or above, this number dropped to 91% for the ONT data but 83% for the HiFi data after phasing, showing the advantage of longer

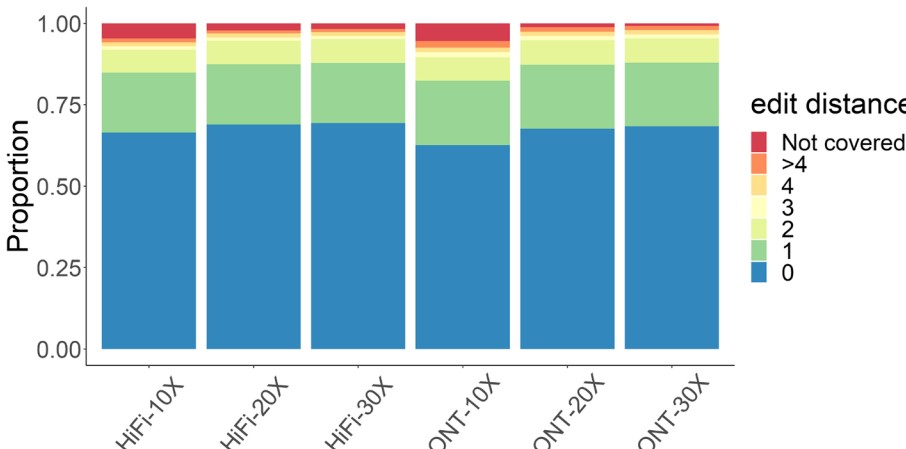

**Fig. 5** Analysis of raw sequencing reads of NA24385. Raw sequencing reads of NA24385 were phased by HapCut2 and annotated by *vamos –read*. Edit distance was calculated by aligning annotation strings to results from the HPRC NA24385 assembly for each VNTR locus

reads for phasing. The HiFi data generally had more loci not covered than the ONT data, though for annotated loci the HiFi data produced slightly better agreement to the assembly annotations. We examined the cases showing small divergence (i.e., edit distance ≤ 2) between the reads and the assembly results and found such loci enriched at regions of low sequence complexities (e.g., AT or GC rich regions), where motifs are highly similar and thus chances of motif misplacement increases.

### Comparison with interval-based merging of population calls

We compared the vamos annotations on the 148 assemblies to a method made to merge variant calls across populations, Jasmine [20]. To generate the variant calls for each assembly, we used the dipcall [34] and applied a filter to exclude variants under 20 bp. The variant calls were then merged across samples using the `jasmine` tool with the `-dup_to_ins` option. Although we excluded smaller variants from analysis, the `jasmine` method was written to use input from a companion program `iris`, which uses variant calls produced by Sniffles [17] and the default setting of Sniffles is to detect variants at least 35 bases making our analysis an overestimate of default behavior. This resulted in a total of 410,418 variants, out of which 153,880 were located in 65,584 VNTRs.

The variants in the Jasmine callset were fully phased, so the distinct alleles could be enumerated based on the pattern of inherited variants each assembly had that overlap with VNTR loci. On average each locus had 5.5 distinct alleles in the Jasmine combined callset when enumerating inherited variants, and 4.0 distinct alleles per locus when aggregating the total gain or loss of variants on each locus. In contrast for vamos, we observed an average of 16.7 alleles per locus when considering distinct annotations, and 7.6 alleles per locus of when only considering the length of each annotation (Table 4). When using the efficient motifs, there was a 15–29% decrease in the number of distinct alleles annotated by vamos, and 1.3–2.6% decrease in the number of length alleles observed (Table 4). On 96% loci, we observed more alleles from the vamos annotation (Fig. 6). This indicates the issue of over-merging in interval-based method, which relies

**Table 4** The average number of alleles per locus obtained from a combined variant callset based on merging of variants by Jasmine, and vamos motif annotations. For the Jasmine calls, the number of alleles was determined using two different definitions. The first definition was based on the presence of known variants in the callset, referred to as "allele by variant" in the table. The second definition was based on the aggregated length of inherited variants, referred to as "allele by total variant length". In contrast, for the vamos method, the table evaluated two definitions of an allele. The first definition was based on the number of motifs in the annotated string, denoted as "allele by length" in the table. The second definition was based on the annotated string of motifs from vamos, denoted as "allele by motif string". The analysis was conducted using the subset of VNTR loci for which Jasmine records a variants ($N = 65{,}584$ for variants $\geq$ 20bp and $N = 46{,}597$ for variants $\geq$ 30bp)

| | Variant >= 20bp | | Variant >= 30bp | |
|---|---|---|---|---|
| Total number of variants | 410,418 | | 288,172 | |
| Total number of variants falling into VNTRs | 153,880 | | 117,431 | |
| Total number of VNTRs intersected with variants | 65,584 | | 46,597 | |
| | # alleles by variant | # alleles by total variant length | # alleles by variant | # alleles by total variant length |
| Interval-based method (Jasmine) | 5.5 | 4.0 | 5.8 | 4.1 |
| vamos | # alleles by motif string | # alleles by length | # alleles by motif string | # alleles by length |
| $q = 0$ | 16.7 | 7.6 | 19.9 | 8.5 |
| $q = 0.1$ | 14.1 | 7.5 | 16.7 | 8.3 |
| $q = 0.2$ | 13.0 | 7.4 | 15.3 | 8.3 |
| $q = 0.3$ | 11.8 | 7.4 | 13.8 | 8.2 |

on heuristics to merge variants based on variant size and breakpoint distance, and can lead to omission of allelic variations at VNTR loci.

## Discussion

New long-read sequencing and assembly efforts are generating large databases of structural variation enriched at variable number of tandem repeats. These variants tend to be multi-allelic, however current catalogs of variation merge multiple different variant calls into a single representative call that limits the ability to associate different alleles with traits. One solution to this is to consider each unique VNTR sequence as an allele. Under this estimate, there are an average of 5.4 alleles per locus in VNTRs among the set of loci examined in this study. Here, we show there is a 24% reduction in annotated allele diversity when small-scale variation is abstracted by the efficient motif set annotation. This difference is subtle, however as the number of genome assemblies increases, the benefit of an abstract encoding of VNTR alleles will also grow, for example in the application of long-read sequencing to association analysis.

The set of regions that vamos calls depends on the initial preprocessing of assemblies. The number of loci annotated in each genome depends on the overall quality of the assembly and the contiguity of the whole-genome alignments between the assembly and the reference. While we were able to partially address missing VNTR loci by including regions masked by TRF in the HGSVC and HPRC assemblies, but missing from GRCh38, future annotations based on the CHM13 T2T assembly [35] will likely

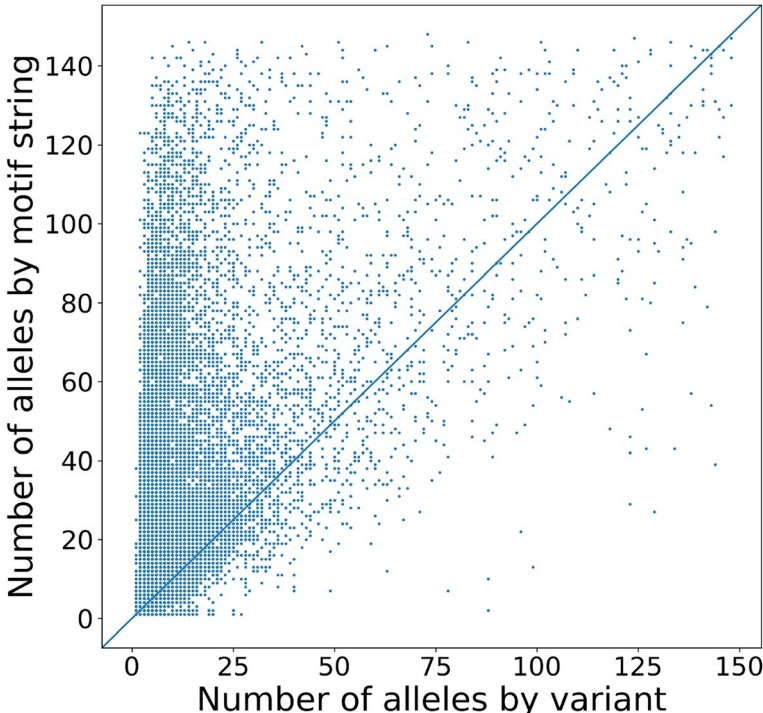

**Fig. 6** Number of alleles comparison between interval-based merging of variants (Jasmine) and motif annotation (vamos). The number of alleles obtained by merging variants (*x*-axis, Jasmine), and by annotating motifs (*y*-axis, vamos). The Jasmine calls reflect a combined callset on variants $\geq 20$ bases, and the vamos annotation using an efficient motif set with $q = 0.1$

increase the number of loci included in the efficient motif database. Additionally, higher quality assemblies will expand the regions where an annotation database may be constructed, such as VNTRs in segmental duplications.

The appropriate parameters for selecting an efficient motif set will be specific to the context which the analysis is performed. Generally, for the reference assemblies that were considered in this study, values of $q \leq 0.1$ preserved common motifs and excluded rare ones. Depending on the aims of the study, a higher value of $q$ will enable characterization of domain structure or other higher-order patterns in repetitive DNA, and low, or complete motif sets can be used to study patterns of rare variation and mutation rates. The use of a standardized motif database (original or efficient) to annotate VNTRs will obscure rare motif variants or repeat interruptions (such as those that may arise in pathogenic disease loci). Future versions of the software may annotate differences between samples and the closest matching motif. Furthermore, it is possible to use custom motif databases when there is a standard representation of motif variation among experts of a particular locus that is not represented by the original TRF annotations. Finally, there is no distinction between removing low frequency motifs that arose from error versus true biological mutations. In this instance, annotations on the original motif set could be retained to be validated by orthogonal sequencing data.

The runtime of ILP solver is highly variable across loci. It takes a few minutes to run for loci with up to hundreds of motifs, but for loci with thousands of motifs, it can

take hours. However, each locus may be calculated in parallel, and it only needs to run once on a set of reference genomes.

## Conclusions

We developed vamos as a tool to automatically curate VNTR motif sets from long-read population assemblies and annotate new genomes using these sets. We show that compared to interval-based merging of variant calls, vamos annotations capture a greater amount of diversity of VNTRs. The method is shown to be robust for annotating on unassembled low-coverage long read data, making it feasible to study large cohorts without high-coverage data and de novo assembly. The vamos software is distributed with efficient motif sets computed for $q = 0.1, 0.2$ and $0.3$, along with the original motif set for the 148 haplotype-assemblies, and may be updated as additional human genomes are assembled.

## Methods

### Generating an original motif set

We used a total of 148 haplotype-resolved assemblies covering nonredundant samples from the HGSVC ($N = 58$) and HPRC ($N = 90$) projects to build a catalog of VNTR motifs. The motif sets were calculated using sequences orthologous to 692,882 VNTR loci defined by the simple repeat track in GRCh38 [36]. Since the GRCh38 VNTR annotations may be missing VNTR annotations due to misassembly and collapsed repeats, we also ran TRF on the 148 assemblies. This identified 5294 additional loci where the lengths on the reference are less than half of the average lengths on assemblies. VNTR loci that are on alternative chromosomes, in centromere regions, longer than 10 kb, or are annotated as more than 40% transposable element were removed and not studied, resulting in a total of 583,316 remaining loci that were considered for analysis.

Given the location of a single VNTR in GRCh38, whole-genome alignments were used to determine the orthologous boundaries across assemblies. The VNTR sequence in the reference and all orthologous VNTR sequences from assemblies are collectively referred to as a single VNTR locus. Each locus was processed first by individual assembly and then collectively across assemblies using TRF, motif filtering, and re-annotation in order to identify their initial motif composition in a manner that is unified across assemblies (Fig. 7A). The VNTR sequences were first extracted from each assembly and annotated in isolation using TRF, where each TRF annotation defines a repeat interval and a consensus motif over that interval. Due to changes in repeat pattern or other interruptions in repeat sequence, TRF may report multiple potentially overlapping annotations or not annotate the entire VNTR sequence. To account for this, the motif composition of each VNTR sequence was re-annotated using the StringDecomposer method with its TRF consensus motifs modified to account for cyclical rotations and filtered to remove homopolymers and redundant motifs that are approximate concatenates of smaller annotations (longer than 1.5 times the length and encompassing 80% a smaller consensus motif). As a result, each VNTR sequence was partitioned into a set of non-overlapping repeat motifs.

A new consensus was generated over the re-annotated motifs by the tool abPOA [37]. Since motifs on the sequence boundaries are often partial, we further eliminated

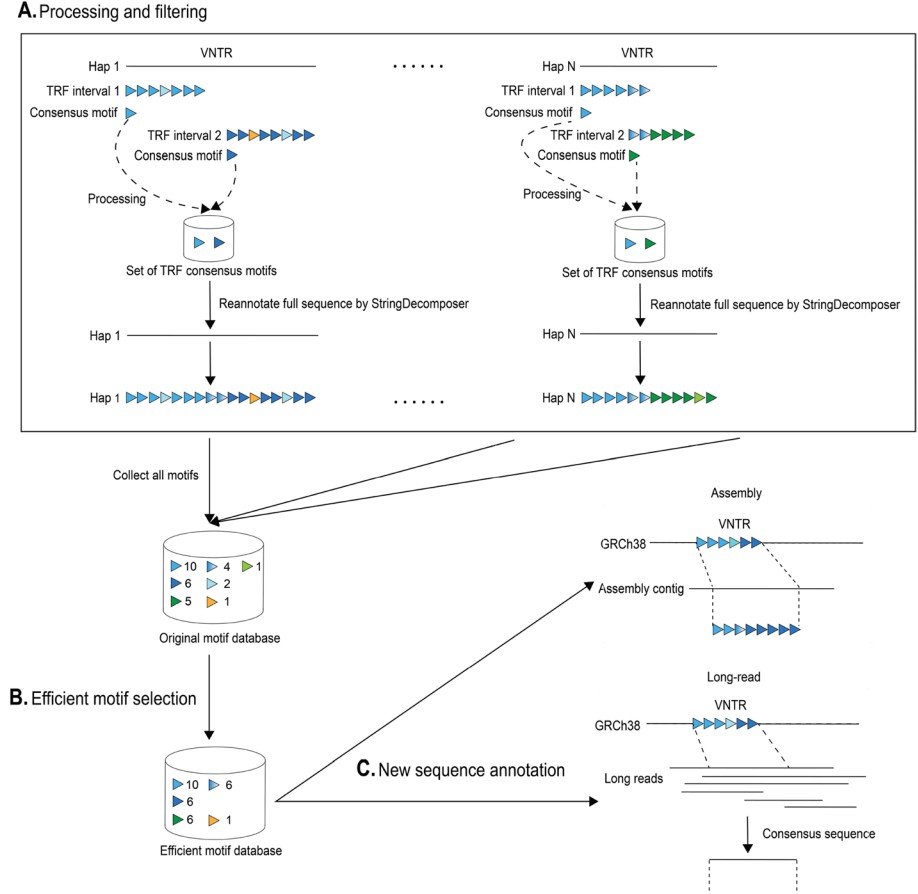

**Fig. 7** vamos workflow of one VNTR locus. **A** Processing and filtering. TRF may report multiple potentially overlapping annotations or not annotate the entire VNTR sequence. So, for each assembly consensus motifs from all TRF annotations are collected to re-annotated the full VNTR sequence by StringDecomposer. Motifs resulted from re-annotation are combined across assemblies as the original motif database. **B** Efficient motif selection. An efficient motif set is selected as a subset of the original motif set under a parameter *q* representing the compression strength. **C** New sequence annotation. New sequences are annotated by StringDecomposer algorithm into a string of efficient motifs that is closest to the raw nucleotide sequence by edit distance

boundary motifs that are shorter than 2/3 or longer than 4/3 of the overall consensus. The full set of reference motifs for a locus were then formed as the non-redundant union of individual sequence motifs from all assemblies, followed by an additional cyclic rotation adjustment so that all motifs are in-frame with the consensus generated by abPOA. To avoid degenerate annotations, homopolymer and dinucleotide loci were excluded ($N = 165{,}308$). Finally, we found that the efficient motif selection and StringDecomposer annotation did not have practical runtimes for loci that had many motifs, and to improve runtime loci that have more than 500 motifs were additionally excluded ($N = 506$).

### Efficient motif selection

The efficient motif selection process is applied independently to each VNTR locus. For a given VNTR locus, let $V = \{v_1, \ldots, v_J\}$ represent a collection of orthologous VNTR sequences from a reference set of assemblies. After processing and filtering in Section

"Generating an original motif set", the set of all distinctly observed motifs in $V$ is referred as original motif set, defined by $\Sigma = \{m_1, m_2, ..., m_p\}$, with count of each motif represented by $o_i$. The efficient motif set is a subset (compressed) from the original motif set (Fig. 7B). During compression, rare motifs are replaced with more common motifs if they are similar, as this suggests that the rare motifs may have mutated from the more common ones or arose from sequencing error. In contrast, rare motifs that are dissimilar to other motifs are retained, as it's less likely that they arose from the mutations of other motifs. The cost of replacing a motif $m_i$ by another motif $m_j$ is defined as the edit distance between $m_i$ and $m_j$, denoted by $\delta_{ij}$. Indicator variable $x_{ij}$ is defined to describe if motif $m_i$ is replaced by $m_j$. We formulate the efficient motif selection problem as an optimization problem. Specifically, we define an efficient motif set $\widetilde{\Sigma} \subseteq \Sigma$ is a minimizer of the weighted sum of the efficient motif set size and the total motif replacement cost. We require the total replacement cost to be bounded by a parameter $\Delta$ to control the compression. We also require if a motif $m_i$ is replaced with motif $m_j$, all occurrences of $m_i$ must be replaced. Furthermore, a motif can be only replaced by another motif with higher or equal count, not the other way around.

The parameter $\Delta$ is specific to each locus, and represents the upper bound of total motif replacement cost. Loci with longer or more divergent motifs require larger $\Delta$ for the efficient motif set to be likely smaller than the original motif set. $\Delta$ is designed to control the number of motifs that can be removed from original motif set and set a reasonable removing cost per motif based on the motif divergence at each locus. Let $M$ be the full list of all observed motifs in $V$. A user-specified global parameter $q \in [0, 1]$ is used to calculate a locus-specific $\Delta$ by setting $\Delta = (Q(q) \times ||M|| * q)$. The value $Q(q)$, represents the $q$-quantile pairwise edit distance of all pairwise edit distances of motifs in $M$ and reflects the motif divergence at a locus, indicating the upper bound of the allowed removing cost per motif. The second term $||M|| * q$ indicates the upper bound of number of motifs that can be removed. The value of $\Delta$ grows with increasing $q$. In the limit, a single motif will be selected for $\widetilde{\Sigma}$ and the efficient motif set will lose the ability to represent variation in composition for a VNTR locus.

Theorem S1 (Additional file 1) guarantees that if efficient motif set exists, the nucleotide sequences translated from vamos annotation string of efficient motifs, differ from the original sequences by at most $\Delta$. Additionally, Theorem S1 (Additional file 1) also indicates that a concatenation of counterpart efficient motif of each original motif in the sequence, referred as *replacement annotation*, is a good approximation of the original raw sequence, with the divergence bounded by $\Delta$. However, it should be noted that theoretically, the vamos annotation is even closer to the original raw sequence.

### Integer linear programming formulation

Theorem S1 (Additional file 1) implies that it is possible to search for an efficient motif set by bounding on the cost of replacing motifs independent of their context in $V$. We prove that the efficient motif set selection problem is a NP-hard problem when the cost function *div* is a general function not limited to edit distance in Theorem S2 (Additional file 2).

Fortunately, it can be formulated as an integer linear programming (ILP) problem (Eq. 2), which can be efficiently solved using the Google OR-tools (CP-SAT solver) [38] .

The problem may be formulated as a linear programming (LP) problem with an indicator function as objective (Eq. 1), which is not yet in ILP form due to the presence of an indicator function in the objective.

$$\min_{x_{ij}} \; \lambda \times \sum_{j=1}^{p} \mathbb{1}(\sum_{i=1}^{p} x_{ij} \geq 1) + \sum_{i=1}^{p} o_i \times \sum_{j=1}^{p} x_{ij}\delta_{ij}$$

$$\text{s.t.} \sum_{i=1}^{p} o_i \times \sum_{j=1}^{p} x_{ij}\delta_{ij} < \Delta$$

$$\sum_{j=1}^{p} x_{ij} = 1, \forall 1 \leq i \leq p$$

$$x_{ij} \times (o_i - o_j) \leq 0, \forall 1 \leq i, j \leq p$$

$$x_{ij} \in \{0, 1\}, \forall 1 \leq i \leq p$$

(1)

However, Eq. 1 can be transformed to an equivalent Eq. 2 by introducing additional variables $y_i = \mathbb{1}(\sum_{i=1}^{p} x_{ij} \geq 1)$. $L$ and $Q$ in Eq. 2 are the lower bound and upper bound of $1 - \sum_{i=1}^{p} x_{ij}$ (In this case, $L$ can be $1 - p$ and $Q$ can be 1). Equation 2 is clearly in ILP form.

$$\min_{y_j} \; \lambda \times \sum_{j=1}^{n} y_j + \sum_{i=1}^{p} o_i * \sum_{j=1}^{p} x_{ij}\delta_{ij}$$

$$\text{s.t.} \sum_{i=1}^{p} o_i * \sum_{j=1}^{p} x_{ij}\delta_{ij} < \Delta$$

$$\sum_{j=1}^{p} x_{ij} = 1, \forall 1 \leq i \leq p$$

$$1 - \sum_{i=1}^{p} x_{ij} \leq Q(1 - y_j), \forall 1 \leq j \leq p$$

$$1 - \sum_{i=1}^{p} x_{ij} \geq (L - 1)y_j + 1, \forall 1 \leq j \leq p$$

$$x_{ij} \times (o_i - o_j) \leq 0, \forall 1 \leq i, j \leq p$$

$$x_{ij} \in \{0, 1\}, \forall 1 \leq i \leq p$$

$$y_j \in \{0, 1\}, \forall 1 \leq j \leq p$$

(2)

### Annotating motif composition on sequence data

The vamos software adapts the StringDecomposer algorithm to annotate the motif composition of tandem repeat sequences for haplotype-resolved assemblies and aligned long reads (for example, when coverage is too low to assemble) (Fig. 7C). As input vamos requires sequence alignments and a file with coordinates of reference VNTRs and a motif list for each VNTR. The tandem repeat sequences to annotate are extracted from the input alignments. When annotating assemblies, sequences are annotated directly from the aligned contigs. When annotating read alignments, all reads covering a VNTR locus are first collected. Reads are partitioned by haplotype using phase tags if they have

been phased using WhatsHap [39] or HapCut2 [40]), or by a max-cut heuristic that initializes partitions with the two reads with the most disagreeing heterozygous SNVs and assigns remaining reads to a partition based on shared SNVs. The VNTR sequences from each of the reads are extracted based on alignments, and the StringDecomposer annotation is executed on the abPOA consensus [37] of the VNTR sequences in each partition.

The output of a vamos run is in Variant Call Format (VCF) [41] with one variant entry per reference VNTR. The efficient motif list for a VNTR is stored in the INFO field along with the vamos annotations of two haplotypes of a sample. Each distinct allele is recorded as a list of motifs from INFO field, and the genotype references which distinct alleles a sample has. In this manner, a combined-sample VCF maintains a record of all observed alleles and enables downstream analysis to compare differences at the level of motif composition.

## Supplementary Information

---

**Additional file 1: Theorem S1** proves that if efficient motif set exists, the replacement annotation is a good approximation of the original raw sequence, with the divergence bounded by Δ.

**Additional file 2: Theorem S2** proves that the efficient motif set selection problem is a NP-hard problem when the cost function *div* is a general function not limited to edit distance.

**Additional file 3.** Review history.

---

### Acknowledgements
We thank Andrey Bzikadze and Pavel Pevzner for their useful comments on StringDecomposer analysis. None of the authors have competing interests.

### Review history
The review history is available as Additional File 3.

### Peer review information

### Authors' contributions
MJPC conceived and designed the study. JR, BG, and MJPC designed the methodological framework. JR, BG, and MJPC implemented the methods, carried out the computational analyses, and drafted the paper. All authors agree to the content of the final paper.

### Funding
This work has been supported by Ming Hsieh Doctoral Fellowship in Computational Biology, the Sloan Foundation, NIH R01HG011649, and NIH 5U24HG007497.

### Availability of data and materials
All the datasets used in this study are publicly available.
• The 64 haplotype-resolved assemblies produced by the Human Genome Structural Variation Consortium [22] are available from http://ftp.1000genomes.ebi.ac.uk/vol1/ftp/data_collections/HGSVC2/release/v1.0/assemblies/ [42].
• The haplotpye-resolved assemblies from the Human Pangenome Reference Consortium [31] are available from https://github.com/human-pangenomics/HPP_Year1_Assemblies/blob/main/assembly_index/Year1_assemblies_v2_genbank.index [43].
• The NA24385 HiFi data is available from NCBI project PRJNA586863 [35].
• The NA24385 ONT Benchmark Datasets was accessed from https://registry.opendata.aws/ont-open-data by the Amazon Web Services S3 bucket at s3://ont-open-data/gm24385_q20_2021.10/.
• Software availability (GPL-2.0 license): https://github.com/chaissonlab/vamos [44] and https://zenodo.org/record/8111620 [45].
• Motif set files and combined vcf of VNTR annotations generated by vamos are available through zenodo at zenodo.org/record/8111620 [45].

## Declarations

### Ethics approval and consent to participate
Not applicable.

**Consent for publication**
Not applicable.

**Competing interests**
The authors declare that they have no competing interests.

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

## 