## [**Additional file 3.** Review history. · Genome Biology]

Review History

First round of review

Reviewer 1

Were you able to assess all statistics in the manuscript, including the appropriateness of statistical tests used? No

Were you able to directly test the methods? No

Comments to author:

Review of "vamos: VNTR annotation using efficient motif sets" by Ren et al. Genome Biology

This manuscript describes a novel algorithm (vamos) that is used to analyze novel genome assemblies generated by PacBio HiFi WGS with the purpose of determining the motif sequence of VNTRs. The algorithm solves a useful, if somewhat niche, problem, in the annotation of the motif sequence of a VNTR is often complex. Current approaches such as Tandem Repeat Finder tend to produce multiple annotations even for a single VNTR locus, and it is not always clear which, if any, of these are actually the unit of variation at any one locus. Vamos potentially provides a better answer than current methods, as it tries to derive a consensus motif based on analyzing multiple PacBio genomes.

While I found the general approach interesting and a potentially useful (but incremental) advance in this specific area of genomics, I was overall not overly impressed with the manuscript, and I thought it fell somewhat short of a standard for publication in Genome Biology. Specifically, the impact of the findings are rather minimal, as the authors simply provide a new set of consensus motifs for VNTRs, and describe how they got them. In itself, for people working in this field this may be useful, but these findings are of limited impact. Also, although providing these new annotations, it's not really clear to me that these are necessarily a big improvement on what exists currently. They MIGHT be, and I suspect they probably are, but there was no demonstration in the manuscript of their utility or how using them makes life better, gives improved results over current VNTR annotations, or informs real biology. So, as a reader, one could not look at this and say that they have clearly come up with something better than already exists, simply that it's an alternative annotation set for VNTRs. The only comparison with current knowledge is a statement of how it results in a different number of "alleles per locus", and in fact I am not really sure the authors even interpret this observation correctly. Again, it was just a statement of difference vs current annotations, without any proof that this is really more accurate or better than if one uses other VNTR annotations. It probably is given how they derived them, but whether that is true or how much better remains to be seen.

In fact, this was something I found lacking, as the manuscript really is just saying "here's our algorithm and here is the resulting data", which is all fine and good, but I was expecting more (ie. a deeper analysis of what happens when you use these and ideally showing how they give marked improvement vs current VNTR annotations) for a publication in this journal. The only real thing the authors do to go beyond this is one line where they compare alleles observed in coding vs non-coding regions, but that was all. In addition, the manuscript overall seemed to have a feel that it was rather scraped together to me, with several of the figures, tables and sections not really fitting together with some seeming to be irrelevant/redundant to the overall story.

Specific comments:

Several of the figures were displayed very small in the PDF I reviewed, which made it very difficult to see detail or read text labels unless zooming in a lot. Please can the authors ensure that submitted figures in any revision use full page widths where possible to make the figures present much more clearly to the reader

Page 3, line 49, states "We generated set of efficient motifs for each VNTR locus under three levels of divergence parameter", but then in Figure 1 it shows and states four divergence parameters were used. Seems like a basic error in description of what was done?

Figure 2, y-axis shows efficient motif set size, ranging from 0.6 to 2.4. What are these units? Above the axis is a small "1e6". Does this mean millions? If so, please use more meaningful and clear annotations so the reader can understand what the plot is showing (ie. if it is millions, please state this clearly)

Page 10, line 14. Authors state "The boundaries of 692,882 VNTR loci were obtained from the UCSC Genome Browser". Please add a definition of what was considered as a VNTR (motif size/copy number?), as it was not stated as far as I could see.

Figure 3 and Table 1 seem rather redundant. Both show compression ratios, but display different quantiles. Wouldn't it be better just to display the figure, and the compression ration could be added as a number above each boxplot? I don't see the table added anything helpful.

Table 2. Unclear what each line is actually displaying, as I do not understand what the inverted "!" means. I am guessing this is some technical symbol, but I have never seen this before, and I am guessing most readers will also not understand it. Please use annotations that are not embedded in computer science jargon. As a result, I am not able to comment on how valuable the data in this table are (or are not) to the manuscript. Another point I found confusing is that the last line of the table is "total", but the sum of the other lines does not add up to anything close to the total. All in all, I found this a very confusing table.... Finally, might they not be better displayed as a figure in any case?

Page 12, line 28. The authors state "Similarly, there is a 42% reduction in the average number of alleles in the efficient motif set under a similar grouping indicating that copy number variation instead of point mutation accounts for the majority of allelic variation". I am not sure that this is true. If I am understanding what the vamos method does correctly, surely if one uses an efficient motif set it will tend to remove/collapse motifs that might differ by a few bp from the consensus. Thus if one sees a reduction in number of alleles with an efficient set, this would indicate the opposite, ie. that many alleles observed are actually similar to each other, simply differing in their internal sequence (ie. point mutations/small indels). Please reconsider if this statement is correct.

Page 13, Section "3.3 Analysis of aligned long-reads". It was unclear why this section is included in the manuscript. It is simply some description of coverage in long read datasets. For me as a reader, it didn't really fit or add to what came prior or after, and I failed to see how it adds anything of relevance to the manuscript, which is focused on description of the vamos algorithm.

Page 13, line 52. Authors state "assemblies. VNTR sequences were simulated based on 500 randomly selected loci described in Section (2.1) and the corresponding replacement annotations were compared with the StringDecomposer annotations" It was unclear to me why only 500 random VNTRs were used, as this only reflects a very tiny proportion of the total of ~418,000 in the genome that they state were used. Surely using more than 0.1% of the dataset would be more robust here? Or is it computationally challenging to expand the set here?

I found Fig4 very hard to decipher, and I thought it did a poor job of getting across the message that I think it is trying to convey. In A and B, the y-axis is listed as log frequency - is this log₁₀, or something else? In C and D, the x-axis goes from 0 to 40, but nearly all the data is in the 0-10 range, meaning everything is tiny and compacted on the left side, and most of the plot is blank space, making it really hard to distinguish coding from non-coding (which is what I think the plot is trying to show?) Also the bars are all contiguous with each other (no white space between pairs), so it's unclear how to compare the distributions of any specific coding to non-coding bar (should I compare with the bar to the left or to the right?). Would it work better to display as a density (line plot) instead of grouped bars? Also in A and B, it's hard to link the lines shown in the plot to which annotation in the legend, in part because the legend uses very small colored lines which look similar. None of this is helped by the fact that each panel is very small.

Figure 5. What does "not annotated" mean? Also I was a little confused, as the authors state in Section 3.3 that using assemblies gives the best performance (not surprising), so it was a bit strange to me that they then present a whole figure of data from analysis of raw reads. Why do this when they already told us raw reads perform worse? (which the plots confirms I think, showing worse annotations correlating with lower coverage). If analyzing assemblies are best, why present data from low coverage raw reads? I am not sure what the conclusion of this plot is, except to say that using low coverage data is worse than using high coverage, which I would think is obvious... Why not just show data from the assemblies if these are best?

Again, Figure 6 I found not very accessible. The figure title says "Relative difference between replacement annotation and StringDecomposer annotation on simulated assemblies", but as far as I can see, the figure does not show this at all (neither of these are annotated anywhere in the figure, which instead displays assembly vs HiFi vs ONT). So, I was not sure the title/legend actually matches the figure shown. And then it's not clear to me what the conclusion or message of the figure is. Sure, the box plots are different with different settings, but what's the point to the reader here in the context of the method?

Reviewer 2

Were you able to assess all statistics in the manuscript, including the appropriateness of statistical tests used? Yes: Statistical tests used were appropriate.

Were you able to directly test the methods? Yes

Comments to author:

The authors describe *vamos*, a toolkit that is used to annotate variable number tandem repeats (VNTRs) in samples. The annotation of the VNTRs relies on building a custom reference

database of VNTRs; briefly, Tandem Repeat Finder (TRF) is used to annotate motifs within a bounded set of coordinates, which are then merged based on a variable q parameter.

Overall, this tool is useful for annotating VNTRs in known polymorphic tandem repeat regions. It is more useful if the authors make their annotated VNTRs available as a resource. It should be noted that since this relies on building a custom "reference database" of motifs per locus beforehand (using orthologous VNTR sequences), it cannot be naively used to annotate tandem repeats that might not be found in the reference genome. In other words, users must know the coordinates and possible motifs in a selected tandem repeat, to annotate the VNTRs within it using this tool. It also has limited use for annotating rare VNTRs or repeat interruptions (such as those that may arise in pathogenic disease loci) unless the q parameter is changed to be very low. In terms of findings, there is limited new biological knowledge gained from this study. Also, the claim of higher number of alleles per VNTR locus largely depends on the parameter setting.

Specific comments:

Figure 1: The figure legend states that "only true representative motifs are selected and rare/contamination motifs possibly resulting from sequence error disappear" - how have the motifs of these samples been validated for every sample? It is unclear how the authors conclude that the motifs that are lost with a higher divergence parameter, are contamination/sequencing error and not true, biological repeat interruptions.

Figure 2: Was divergence also tested with the same q parameters as in Fig. 1 (0.25 and 0.35)? Minor point: Would be best to keep percentage/parameter convention the same (i.e. 0.1 vs. 10%) to increase clarity.

Page 11: The authors discuss an example in which the greedy approach and the efficient approach report distinct motifs when annotating the VNTR. Although the efficient approach reports the (potentially) rare motif, would the greedy algorithm not report the more biologically relevant motif here? Was there any more work done on investigating the biological repercussion of different VNTRs at any single locus, to determine whether important data is lost through either approach?

Page 12: They suggest that the lack of motif diversity between coding and noncoding VNTR could be due to a constraint on mutation in coding sequences. However, it cannot explain why it is not different from noncoding sequences, where constraint is reduced.

Page 13: The authors mention that efficient motif annotation reports 63 alleles for the WDR7 locus while the HGVSC call set reports less. A "truth" set would help answer which is more accurate here. Does changing the q parameter more accurately reflect the call-set data (i.e. changing to a q of 0.1)? Also, how does the greedy algorithm perform on this?

Minor comment:

Figure 4: The authors might consider limiting the x-axis of graphs C and D to make the distribution of coding vs. non-coding motifs clearer. The number of alleles does not equal the number of motifs, so the graphs from A/B do not need to have the same x-axis limits as C/D.

We would like to begin our response with a description of a change in input where additional assemblies from the Human Pangenome Reference Consortium (HPRC) were included in the construction of our reference motif set. This dataset became available after initial submission of this manuscript. Because we would want a reference motif set that reflects as many individuals as possible, we would have included this dataset in the motif sets distributed with vamos. We have elected to include this dataset in the revision of our manuscript so that the motif sets released with the software are consistent with the publication, however the change of source data for our efficient motif sets necessarily changed most of the statistics in the manuscript. By adding additional assemblies we were able to increase the total diversity of motifs that we sample, and so most of the statistics reflect this increase in diversity. The overall conclusion, that it is necessary to annotate VNTR sequences by motif composition rather than by merging variant calls, remains the same (in fact increases).

Furthermore, when preparing a separate publication, we found that many of the TRF annotations in GRCh38 included mobile elements that happened to be in tandem. These did not show any copy number variation, and were removed from analysis here.

Below is the list of major changes to the current version of the manuscript in order to accomplish this change:

1. Added the HPRC genomes to increase the number of haplotypes from 64 to 148
2. Added 5294 novel VNTRs that are not on GRCh38.
3. Changed the preprocessing procedures to include full sequence features of VNTR sequences and filter transposable elements (Alu/SINE/LINE).
4. Compared vamos annotations of 148 haplotypes with the allele aggregation by an interval-based approach.
5. Switched major analysis from $q=0.2$ to $q=0.1$ to highlight a more diverse set of alleles.
6. Increased sample size of simulation from 500 to 50,000.
7. Deleted the analysis about coding and non-coding regions.

To improve readability, we have largely rewritten some of the introduction and most of Sections 2.1, and 2.3.

Reviewer #1: Review of "vamos: VNTR annotation using efficient motif sets" by Ren et al.
Genome Biology

This manuscript describes a novel algorithm (vamos) that is used to analyze novel genome assemblies generated by PacBio HiFi WGS with the purpose of determining the motif sequence of VNTRs. The algorithm solves a useful, if somewhat niche, problem, in the annotation of the motif sequence of a VNTR is often complex. Current approaches such as Tandem Repeat Finder tend to produce multiple annotations even for a single VNTR locus, and it is not always clear

which, if any, of these are actually the unit of variation at any one locus. Vamos potentially provides a better answer than current methods, as it tries to derive a consensus motif based on analyzing multiple PacBio genomes.

While I found the general approach interesting and a potentially useful (but incremental) advance in this specific area of genomics, I was overall not overly impressed with the manuscript, and I thought it fell somewhat short of a standard for publication in Genome Biology. Specifically, the impact of the findings are rather minimal, as the authors simply provide a new set of consensus motifs for VNTRs, and describe how they got them. In itself, for people working in this field this may be useful, but these findings are of limited impact. Also, although providing these new annotations, it's not really clear to me that these are necessarily a big improvement on what exists currently. They MIGHT be, and I suspect they probably are, but there was no demonstration in the manuscript of their utility or how using them makes life better, gives improved results over current VNTR annotations, or informs real biology. So, as a reader, one could not look at this and say that they have clearly come up with something better than already exists, simply that it's an alternative annotation set for VNTRs. The only comparison with current knowledge is a statement of how it results in a different number of "alleles per locus", and in fact I am not really sure the authors even interpret this observation correctly. Again, it was just a statement of difference vs current annotations, without any proof that this is really more accurate or better than if one uses other VNTR annotations. It probably is given how they derived them, but whether that is true or how much better remains to be seen.

In fact, this was something I found lacking, as the manuscript really is just saying "here's our algorithm and here is the resulting data", which is all fine and good, but I was expecting more (ie. a deeper analysis of what happens when you use these and ideally showing how they give marked improvement vs current VNTR annotations) for a publication in this journal. The only real thing the authors do to go beyond this is one line where they compare alleles observed in coding vs non-coding regions, but that was all. In addition, the manuscript overall seemed to have a feel that it was rather scraped together to me, with several of the figures, tables and sections not really fitting together with some seeming to be irrelevant/redundant to the overall story.

We acknowledge the reviewer's concerns about the work presented here being incremental. To refute that, we point out that the majority of variation discovered by long reads is in VNTR sequences, and this type of variation has been missed by most short-read studies in the past (Zhao, X et al. AJHG, 2021). Thus, the long-read studies of large populations that are under way: TOPMed, 1kg-ONT, and CARD (to name those that we are aware of), will provide their novelty in ~400 "challenging genes" in repetitive DNA missed by short-read sequencing (Wagner, J., et al., Nature Biotechnology, 2022), and previously missed variation in VNTR sequences.

We have included extended analysis comparing the annotations by our method to those of a method recently published in Nature Methods, Jasmine (Kirsche et al., 2023). We show that if researchers use this method to combine calls across individuals, there is a massive underrepresentation in diversity of VNTR sequences in the combined callset.

This is a forward-looking method in anticipation of these larger datasets. For now, the increased diversity is the major point of the manuscript, and with the larger and more deeply phenotyped cohorts being sequenced, the full functional impact of VNTR variation will be more easily measured using vamos.

Specific comments:

Several of the figures were displayed very small in the PDF I reviewed, which made it very difficult to see detail or read text labels unless zooming in a lot. Please can the authors ensure that submitted figures in any revision use full page widths where possible to make the figures present much more clearly to the reader

We have made figures use full page width. If the manuscript is accepted, we will work with the publisher to ensure all text labels are readable.

Page 3, line 49, states "We generated set of efficient motifs for each VNTR locus under three levels of divergence parameter", but then in Figure 1 it shows and states four divergence parameters were used. Seems like a basic error in description of what was done?

There are three parameters for divergence greater than 0 (original set), and the original set. We have updated the text to clarify this:

We generated an efficient motif set for VNTR loci from 148 haplotype-resolved assemblies sequenced with LRS by the Human Genome Structural Variant Consortium (HGSVC) (Ebert et al., 2021) and the Human Pangenome Reference Consortium (HPRC) (Liao et al., 2022) and under three levels of divergence, as well as the original motifs.

Figure 2, y-axis shows efficient motif set size, ranging from 0.6 to 2.4. What are these units? Above the axis is a small "1e6". Does this mean millions? If so, please use more meaningful and clear annotations so the reader can understand what the plot is showing (ie. if it is millions, please state this clearly)

We have updated the figure axis to clarify M.

Page 10, line 14. Authors state "The boundaries of 692,882 VNTR loci were obtained from the UCSC Genome Browser". Please add a definition of what was considered as a VNTR (motif size/copy number?), as it was not stated as far as I could see.

We have moved this statement to the “Materials and Methods” section. We used tandem repeat annotations as defined by the simple repeats track on the UCSC genome browser. We have updated this text to:

The motif sets were calculated using sequences orthologous to 692,882 VNTR loci defined by the simple repeat track in GRCh38 (Kent 2008).

Figure 3 and Table 1 seem rather redundant. Both show compression ratios, but display different quantiles. Wouldn't it be better just to display the figure, and the compression ration could be added as a number above each boxplot? I don't see the table added anything helpful.

We have removed Figure 3 and only kept Table 1.

Table 2. Unclear what each line is actually displaying, as I do not understand what the inverted "!" means. I am guessing this is some technical symbol, but I have never seen this before, and I am guessing most readers will also not understand it. Please use annotations that are not embedded in computer science jargon. As a result, I am not able to comment on how valuable the data in this table are (or are not) to the manuscript. Another point I found confusing is that the last line of the table is "total", but the sum of the other lines does not add up to anything close to the total. All in all, I found this a very confusing table.... Finally, might they not be better displayed as a figure in any case?

The ! was an invalid conversion of “<” character. This table is to compare two annotation motif databases, motif sets generated by vamos and the greedy approach. Each line represents the average number of loci that one method is better or significantly better than the other. We have revised the text to reflect this using:

Comparison between vamos and greedy method. The quality of an annotation was measured calculating the edit distance between the nucleotide sequences translated from the annotated string of motifs and the original assembled sequences. An annotation is significantly better if the corresponding edit distance is less than 80% of that of the other

Page 12, line 28. The authors state "Similarly, there is a 42% reduction in the average number of alleles in the efficient motif set under a similar grouping indicating that copy number variation instead of point mutation accounts for the majority of allelic variation". I am not sure that this is true. If I am understanding what the vamos method does correctly, surely if one uses an efficient motif set it will tend to remove/collapse motifs that might differ by a few bp from the consensus. Thus if one sees a reduction in number of alleles with an efficient set, this would indicate the opposite, ie. that many alleles observed are actually similar to each other, simply

differing in their internal sequence (ie. point mutations/small indels). Please reconsider if this statement is correct.

The reduction in number of alleles in the original set would include both those from point mutation and length variation, whereas the reduction in number of alleles in the annotations by efficient motifs would mostly include length variation because rare motifs are already removed. We have reworded this section to be more clear, specifically:

When considering annotations using the original motif set, there was a 47% reduction in the average number of alleles per locus when grouping alleles that differ by up to two motifs by an edit distance allowing for insertion and deletion in addition to motif mismatch. Similarly, there was a 44% reduction in the average number of alleles in the efficient motif set under a similar grouping by edit distance. Because the annotation by efficient motif sets already have a reduced diversity due to motif compression, the similar relative reduction by edit distance in both annotations indicates relatively few motif mismatches and that copy number variation instead accounts for the majority of allelic variation, as expected by the slippage mutational mechanism of tandem repeat sequences.

Note, the reduction in the efficient motif annotation has changed due to using the HGVC+HPRC genomes.

Page 13, Section "3.3 Analysis of aligned long-read". It was unclear why this section is included in the manuscript. It is simply some description of coverage in long read datasets. For me as a reader, it didn't really fit or add to what came prior or after, and I failed to see how it adds anything of relevance to the manuscript, which is focused on description of the vamos algorithm.

Not all new long-read projects are guaranteed to generate sufficient coverage for assembling genomes. In fact, the NIH All of Us pilot study did comprehensive analysis of sensitivity at low coverage, presumably so that more samples could be sequenced with the same level of funding (<https://www.biorxiv.org/content/10.1101/2023.01.23.525236v1.full.pdf>). The type of association analysis that vamos enables will require large datasets, and this analysis of the effectiveness of VNTR calling from low-coverage sequencing helps researchers design their studies accordingly. To emphasize this, we have modified the sentence:

We analyzed long-read data of NA24385 sequenced at 10-30X for both HiFi and ONT platforms

To:

Because studies may elect to perform low-coverage sequencing to increase sample size, we analyzed long-read data of NA24385 sampled at 10-30X for both HiFi and ONT platforms to investigate the accuracy of VNTR annotation on real mapped reads.

Page 13, line 52. Authors state "assemblies. VNTR sequences were simulated based on 500 randomly selected loci described in Section (2.1) and the corresponding replacement annotations were compared with the StringDecomposer annotations" It was unclear to me why only 500 random VNTRs were used, as this only reflects a very tiny proportion of the total of ~418,000 in the genome that they state were used. Surely using more than 0.1% of the dataset would be more robust here? Or is it computationally challenging to expand the set here?

We expanded analysis to 50,000 loci, excluding loci with a single motif because such analysis is trivial. The overall conclusion is similar, but the relative difference of each group increased overall. This was not due to use of more loci but the update of motif set. The updated motif set is generally more diverse because of changes in preprocessing, decreased condensation strength (from $q=0.2$ to $q=0.1$), and inclusion of the HPRC samples. Similar increase of mean relative difference is also observed with only 500 loci using the updated motif set. The average increase of mean relative difference of 500 random loci was 4% compared to the old data.

I found Fig4 very hard to decipher, and I thought it did a poor job of getting across the message that I think it is trying to convey. In A and B, the y-axis is listed as log frequency - is this log10, or something else? In C and D, the x-axis goes from 0 to 40, but nearly all the data is in the 0-10 range, meaning everything is tiny and compacted on the left side, and most of the plot is blank space, making it really hard to distinguish coding from non-coding (which is what I think the plot is trying to show?) Also the bars are all contiguous with each other (no white space between pairs), so it's unclear how to compare the distributions of any specific coding to non-coding bar (should I compare with the bar to the left or to the right?). Would it work better to display as a density (line plot) instead of grouped bars? Also in A and B, it's hard to link the lines shown in the plot to which annotation in the legend, in part because the legend uses very small colored lines

which look similar. None of this is helped by the fact that each panel is very small.

We have removed Figure4C and Figure4D from the manuscript. We have also combined Figure4A and Figure4B into a single plot to make comparisons of the efficient groups to the original groups easier. The y-axis has been noted as log10 scale and clipped for more efficient space use. We also removed the $ed=1$ groups as they show trivial results. The HGSVC variant call set has been combined with the HPRC call set and analyzed by the interval-based merging method (Jasmine and Iris) in section 3.5. Thicker lines and more divergent colors have been used for different lines and the legend.

Figure 5. What does "not annotated" mean? Also I was a little confused, as the authors state in Section 3.3 that using assemblies gives the best performance (not surprising), so it was a bit strange to me that they then present a whole figure of data from analysis of raw reads. Why do this when they already told us raw reads perform worse? (which the plots confirms I think, showing worse annotations correlating with lower coverage). If analyzing assemblies are best, why present data from low coverage raw reads? I am not sure what the conclusion of this plot is, except to say that using low coverage data is worse than using high coverage, which I would think is obvious... Why not just show data from the assemblies if these are best?

We have revised the analysis to state "Not covered". In the initial submission, "not annotated" referred to both not covered, and not phased loci. In this revision, we treat not phased as autozygous. To address this, we have added the sentence to Section 3.3:

Regions where reads could not be phased, but still covered a VNTR locus were annotated with homozygous calls.

Again, Figure 6 I found not very accessible. The figure title says "Relative difference between replacement annotation and StringDecomposer annotation on simulated assemblies", but as far as I can see, the figure does not show this at all (neither of these are annotated anywhere in the figure, which instead displays assembly vs HiFi vs ONT). So, I was not sure the title/legend actually matches the figure shown. And then it's not clear to me what the conclusion or message of the figure is. Sure, the box plots are different with different settings, but what's the point to the reader here in the context of the method?

The caption has been rewritten to be more clear; we do not use the term replacement annotation because this was unclear. The figure shows the difference between variant annotations and the annotations derived by replacing original motifs with their counterpart efficient motifs.

Reviewer #2: The authors describe vamos, a toolkit that is used to annotate variable number tandem repeats (VNTRs) in samples. The annotation of the VNTRs relies on building a custom reference database of VNTRs; briefly, Tandem Repeat Finder (TRF) is used to annotate motifs within a bounded set of coordinates, which are then merged based on a variable q parameter.

Overall, this tool is useful for annotating VNTRs in known polymorphic tandem repeat regions. It is more useful if the authors make their annotated VNTRs available as a resource.

We note that VCFs generated as part of this manuscript are available for download from Zenodo at the following link (updated from the original submission because of new content):

<https://zenodo.org/record/7884547#.ZFA8S9LMI28>

Furthermore, running *vamos* users are presented with links to download the most recent motif sets.

It should be noted that since this relies on building a custom "reference database" of motifs per locus beforehand (using orthologous VNTR sequences), it cannot be naively used to annotate tandem repeats that might not be found in the reference genome. In other words, users must know the coordinates and possible motifs in a selected tandem repeat, to annotate the VNTRs within it using this tool.

This is correct, *vamos* is not a de novo VNTR discovery tool. To address deficiencies in VNTR annotations in GRCh38, we added an additional 5,294 loci that were discovered using Tandem Repeats Finder on genomes assembled by the Human Pangenome Reference Consortium and Human Genome Structural Variation Consortium. While it may be possible for small groups of individuals to have a repeat expansion not found elsewhere in the population, we anticipate the current annotation database covers most of the VNTR sequences in the human genome. We also state:

...future annotations based on the CHM13 T2T assembly (Nurk et al. 2022) will likely increase the number of loci included in the efficient motif database.

It also has limited use for annotating rare VNTRs or repeat interruptions (such as those that may arise in pathogenic disease loci) unless the *q* parameter is changed to be very low.

We agree, and view *vamos* as a tool to enable association analysis of more common motif variation. We are including the reviewer statement in the manuscript as a critique in the discussion, along with a forward looking statement for the potential to keep track of differences between samples and motif databases:

The use of a standardized motif database (original or efficient) to annotate VNTRs will obscure rare motif variants or repeat interruptions (such as those that may arise in pathogenic disease loci). Future versions of the software may annotate differences between samples and the closest matching motif.

In terms of findings, there is limited new biological knowledge gained from this study. Also, the claim of higher number of alleles per VNTR locus largely depends on the parameter setting.

We feel that the datasets that will take advantage of this tool are just now being generated in large-scale consortia such as TOPMed, 1KG-ONT, and the CARD consortium, each of which is sequencing thousands of individuals using long-read data of various types. The discussion of alleles per locus, while seemingly nuanced, importantly highlights how much current analysis methods obscure the true level of variation in these loci.

To address this, we have added an additional section “Comparison with interval-based merging of population calls”, where we compared the diversity measured using vamos to the diversity measured using a tool recently published in Nature Methods, Jasmine (Kirsche et al., 2023). This approach uses the “traditional” clustering of variants using breakpoint similarities. This vastly understates the allelic diversity of VNTR sequences. The figure below (now included in the manuscript) shows how highly diverse alleles annotated by vamos are collapsed into a much smaller fraction by vamos.

Specific comments:

Figure 1: The figure legend states that "only true representative motifs are selected and rare/contamination motifs possibly resulting from sequence error disappear" - how have the motifs of these samples been validated for every sample? It is unclear how the authors conclude that the motifs that are lost with a higher divergence parameter, are contamination/sequencing error and not true, biological repeat interruptions.

This is a good point. Without orthogonal data (e.g. duplex Nanopore sequencing data), we cannot distinguish between errors and repeat interruptions. We address this in the discussion:

Finally, there is no distinction between removing low frequency motifs that arose from error versus true biological mutations. In this instance, annotations on the original motif set could be retained to be validated by orthogonal sequencing data.

Figure 2: Was divergence also tested with the same q parameters as in Fig. 1 (0.25 and 0.35)? Minor point: Would be best to keep percentage/parameter convention the same (i.e. 0.1 vs. 10%) to increase clarity.

We have modified Figure 1 (now Figure 2) so that it uses the same parameters as the rest of the manuscript.

Page 11: The authors discuss an example in which the greedy approach and the efficient approach report distinct motifs when annotating the VNTR. Although the efficient approach reports the (potentially) rare motif, would the greedy algorithm not report the more biologically relevant motif here? Was there any more work done on investigating the biological repercussion of different VNTRs at any single locus, to determine whether important data is lost through either approach?

While our illustrative example shows a more rare motif, we measured how similar the annotated sequences are to the original sequence, for both greedy and efficient motif sets. We show that at loci with at least 5 motifs (those likely to show a difference between approaches for selecting a reduced motif set), the efficient motif set reflects a closer sequence to the original sample VNTR at 12.5% more loci than the greedy annotation, and a 28.3% relative increase for loci where the difference is considerable (where one annotation has at least 20% more differences than the other). While this does not answer the question about biological repercussions, it shows that, on average, the efficient motif set has a more faithful representation of the original loci than reduced.

Page 12: They suggest that the lack of motif diversity between coding and noncoding VNTR could be due to a constraint on mutation in coding sequences. However, it cannot explain why it is not different from noncoding sequences, where constraint is reduced.

We agree, and have removed this analysis.

Page 13: The authors mention that efficient motif annotation reports 63 alleles for the WDR7 locus while the HGVSC call set reports less. A "truth" set would help answer which is more accurate here. Does changing the q parameter more accurately reflect the call-set data (i.e. changing to a q of 0.1)? Also, how does the greedy algorithm perform on this?

The interval based merging of variants used by the HGSVC, by construction, merges together sequences that are different into the same variant, assuming the differences are below some threshold defined by the implementation of the merging method. In this instance, truth may be considered the original sequences (assuming the assemblies are error-free), and the number of different alleles as the number of exactly differing sequences. The relative difference between the number of alleles reported by vamos or different annotation method (e.g. HGSVC or

Jasmine), and the ground truth established by sequence differences reflects the degree to which true variation is obscured by a variant annotation or merging approach. Note, we previously used the HGSVC dataset, and now use Jasmine to merge variants in order to reflect a larger dataset.

We can use the vamos annotations with $q=0$ (original motif sequences) as a proxy for the annotations that reflect the total number of alleles at a locus. In Table 4, we show that among the VNTR loci where a variant considered by Jasmine was in a VNTR ($N=65,584$), there is a 66% reduction in allelic diversity using Jasmine merging versus a 15% reduction in allelic diversity using vamos efficient motifs ($q=0.1$). The greedy algorithm would likely have a similar reduction in diversity, however the extent to which rare alleles are replaced by more common ones depends on the specific application.

Minor comment:

Figure 4: The authors might consider limiting the x-axis of graphs C and D to make the distribution of coding vs. non-coding motifs clearer. The number of alleles does not equal the number of motifs, so the graphs from A/B do not need to have the same x-axis limits as C/D.

We have removed this plot, as this analysis was not a major contribution to the take-home message of the manuscript.

Second round of review

Reviewer 1

I think the authors have answered the point raised in initial review reasonably well. There are still a couple of minor niggles (eg Fig2 they just put "M" to mean million, which is not a standard abbreviation nor is it explained anywhere), but I think this can be sorted out during copy editing

Reviewer 2

The authors have satisfactorily responded to my previous comments.

Reviewer #1: I think the authors have answered the point raised in initial review reasonably well. There are still a couple of minor niggles (eg Fig2 they just put "M" to mean million, which is not a standard abbreviation nor is it explained anywhere), but I think this can be sorted out during copy editing

We've modified the figure by labeling the numerical unit explicitly as "millions" on the y-axis. The new figure is attached here.

Furthermore, we found that the indicator variable "xij" was defined in the additional file but not in the main text. So, we added the following sentence at Section 5.2 (line 400) to define the term.

Indicator variable xij is defined to describe if motif mi is replaced by mj.

In addition to the above changes, we've also corrected the following typos in the manuscript.

Section 2.1 (line 143)

...were ran on the 48 HPRC and HGSVC assemblies.

changed to

...were ran on the 148 HPRC and HGSVC assemblies.

Table 3

top-left cell with value 4.9 (7.8)
changed to
4.8 (7.8)

Section 2.2 (line 216)

...under the original motif set was 4.9 (7.8 excluding constant loci)
changed to
...under the original motif set was 4.8 (7.8 excluding constant loci)

Section 2.2 (line 223)

...there was a 47% reduction in the average number of alleles per locus...
changed to
...there was a 46% reduction in the average number of alleles per locus...

Section 2.3 (line 257)

...annotations of new genomes not included the reference assemblies...
changed to
...annotations of new genomes not included in the reference assemblies...

Section 2.4 (line 277)

...leads to better phasing and construction of more accurate consensus sequences for improved overall performance at higher sequencing depth.
changed to
...leads to better phasing and construction of more accurate consensus sequences for improved overall performance.

Section 2.5 (line 303)

...and can lead to omission allelic variations at VNTR loci.
changed to
...and can lead to omission of allelic variations at VNTR loci.

Section 3

First paragraph and the last sentence of Section 3 (Discussion) was moved to Section 4 (Conclusion)

Section 5.2 (line 398)

After processing and filtering in section 5.1
changed to
After processing and filtering in Section 5.1

Section 5.2 (line 400)

The cost of replacing a motif m_j by another motif m_i is defined as...
changed to
The cost of replacing a motif m_i by another motif m_j is defined as...

Section 5.2 (line 404)

We also require if a motif m_j is replaced with motif m_i ,
changed to

We also require if a motif m_i is replaced with motif m_j ,

Equation 1 (line 430)

$x_{ij} \times (o_i - o_j) \geq 0, \forall 1 \leq i, j \leq p$

changed to

$x_{ij} \times (o_i - o_j) \leq 0, \forall 1 \leq i, j \leq p$

Reference of Theorem

Theorem 6.1 (Appendix)

changed to

Theorem S1 (Additional file 1)

Reference of Theorem

Theorem 6.2 (Appendix)

changed to

Theorem S2 (Additional file 2)

Theorem S1 (additional_file_S1 line 44)

$x_{ij} * \delta_{ij} \leq \Delta, v_j \in V$

changed to

$x_{ij} * \delta_{ij} < \Delta, v_j \in V$

Theorem S1 (additional_file_S1 line 48)

Motif m_i cannot be replaced by motif m_j if $o_i > o_j$.

changed to

Motif m_i cannot be replaced by motif m_j if $o_i \geq o_j$.

Theorem S2 (additional_file_S2 line 48)

the second requirement in 5.2 will always be satisfied.

changed to

the second requirement in Theorem S1 (Additional file 1) will always be satisfied.

Theorem S2 (additional_file_S2 line 57)

$\delta_{ij} x_{ij} \leq \Delta$ in polynomial time.

changed to

$\delta_{ij} x_{ij} < \Delta$ in polynomial time.

Theorem S2 (additional_file_S2 line 67)

$\delta_{ij} x_{ij} \leq \Delta$ in polynomial time.

changed to

$\delta_{ij} x_{ij} < \Delta$ in polynomial time.

Theorem S2 (additional_file_S2 line 84)

$0 \leq \Delta$

changed to

$0 < \Delta$